# Clinical value of metagenomic next-generation sequencing in screening oropharyngeal colonization in patients undergoing allogeneic hematopoietic stem cell transplantation: a prospective observational study

Yuqi Zhang,[1,2,3] Jianrong Ge,[1,4] Yuhang Wang,[1,2,3] Zaixiang Tang,[5] Xiao Ma,[1,2,3] JIsheng Liu,[1,4] Depei Wu,[1,2,3] Xiaojin Wu[1,2,3]

**ABSTRACT** Screening for colonization is an essential procedure in allogeneic hematopoietic stem cell transplantation (allo-HSCT). Although metagenomic next-generation sequencing (mNGS) has played an important role in the diagnosis of complex and challenging infections, its effectiveness in screening oropharyngeal colonization is not yet fully assessed. We performed a prospective analysis (ChiCTR2300069450) involving 128 allo-HSCT patients between June 2022 and June 2023. Before the conditioning regimen, all patients underwent oropharyngeal and anal swab tests to detect colonizing pathogens. In addition to culture-based methods, we also analyzed oropharyngeal swab samples using mNGS. Among the allo-HSCT patients, the overall colonization rate from cultures was 15.6%, while mNGS identified an oropharyngeal colonization rate of 49.2%. Patients with oropharyngeal Enterobacteriaceae colonization had a higher incidence of post-transplant bloodstream infection (BSI) (39.1% vs 19.0%, $P = 0.034$) and thrombotic microangiopathy (17.4% vs 8.5%, $P = 0.04$). Multivariate analysis confirmed oropharyngeal Enterobacteriaceae colonization as an independent risk factor for non-relapse mortality (NRM), overall survival (OS), and progression-free survival (PFS) ($P = 0.024, 0.030,$ and $0.021$, respectively). The individuals with carbapenem-resistant Enterobacteriaceae (CRE) colonization experienced delayed platelet engraftment ($P = 0.018$). Moreover, they had significantly worse OS ($P = 0.002$), higher NRM ($P = 0.00015$), and poorer PFS ($P = 0.00095$). Screening for oropharyngeal colonization using mNGS provides critical clinical value in predicting transplant prognosis. Clinicians should closely monitor patients with oropharyngeal Enterobacteriaceae or CRE colonization.

**IMPORTANCE** Screening for colonization is essential for predicting infection risk in allo-HSCT patients. Traditional microbiological testing methods, however, are time-consuming and have low sensitivity. In this paper, we examine the impact of oropharyngeal colonization on outcomes following allo-HSCT while also evaluating the utility of mNGS for detecting colonization. Our investigation reveals that screening for oropharyngeal colonization using mNGS provides critical clinical value in predicting transplant outcomes and prognosis. Additionally, not all colonization has clinical relevance, but oropharyngeal Enterobacteriaceae colonization has negative impacts on transplant prognosis. Colonization by CRE had particularly severe consequences, which warrants serious attention.

**CLINICAL TRIALS** This study is registered as a single-center clinical trial (Registration No. ChiCTR2300069450).

**Peer Reviewer** Brian Scott Grundy, University of Virginia, Charlottesville, Virginia, USA

Address correspondence to Xiaojin Wu, wuxiaojin@suda.edu.cn.

Yuqi Zhang, Jianrong Ge, and Yuhang Wang contributed equally to this article. The author order was determined by drawing straws.

The authors declare no conflict of interest.

See the funding table on p. 13.

**KEYWORDS** metagenomic next-generation sequencing, hematopoietic stem cell transplantation, colonization, Enterobacteriaceae, prognosis

Patients undergoing allogeneic hematopoietic stem cell transplantation (allo-HSCT) frequently experience neutropenia and lymphocyte depletion due to high-intensity conditioning regimens. This condition leaves the body in a prolonged state of immuno-suppression or immune deficiency, disrupting the balance of its microbial ecosystem. The oral cavity, second only to the gastrointestinal tract, hosts a diverse microbial community comprising over 700 bacterial species adhering to the oral mucosa, most of which are non-pathogenic commensals under normal conditions (1, 2). However, in the context of allo-HSCT, opportunistic pathogens in the oropharynx can become dominant. These pathogens may spread to the lower respiratory and gastrointestinal tracts, further disrupting microbial balance and increasing the risk of various infections (3). Moreover, high-intensity conditioning regimens elevate the incidence of chemotherapy-associated oropharyngeal mucositis. This condition allows opportunistic pathogens to enter the bloodstream directly through the compromised mucosal barrier, resulting in bloodstream infections (BSIs) (4, 5). Previous studies indicate that patients with hematologic malignancies or solid tumors have a significantly higher oral colonization rate of Gram-negative bacteria (GNB) compared to healthy individuals, with *Escherichia coli* being the most commonly isolated bacterium among these patients, potentially heightening infection (6). A prior study conducted at our center employed culture methods to screen for oropharyngeal colonization in 1,270 allo-HSCT patients, revealing a pathogen colonization rate of approximately 10%. *Klebsiella pneumoniae* and *Enterobacter cloacae* ranked among the most frequently detected Enterobacteriaceae species (7). The association between the oropharyngeal microbiota and the relapse as well as graft-versus-host disease (GVHD) after transplantation has been identified by previous studies (8, 9). Therefore, investigating oropharyngeal microbial colonization in allo-HSCT patients is of significant clinical importance.

Identifying colonization through culture can take several days, which presents practical limitations in isolating and characterizing the number and diversity of species within complex microbial communities. Kropshofer G demonstrated that culture-based techniques for monitoring microbial colonization in pediatric hematopoietic stem cell transplantation provide no significant benefit (10). In contrast, metagenomic next-generation sequencing (mNGS) technology, known for its rapid speed and high throughput, is increasingly applied to various genomic and transcriptomic sequencing task (11). Because nearly all microorganisms contain DNA or RNA genomes, mNGS serves as an efficient tool for investigating a broad spectrum of pathogens (12). Compared to traditional detection methods, mNGS offers higher sensitivity in pathogen identification and is less influenced by prior antibiotic exposure (13). Currently, in clinical practice, mNGS is often used as an adjunct to routine culture, antibody, antigen, and/or PCR detection. However, despite its advantages, mNGS also presents significant challenges, acting as a "double-edged sword." Key limitations of this technology include significantly increased costs compared to routine methods, specialized trained personnel, difficulties in distinguishing colonization from infection, the presence of exogenous nucleic acids, and the lack of standardized methodologies. Consequently, the role of mNGS in detecting colonization remains poorly understood.

This study aims to evaluate the utility of mNGS in screening for oropharyngeal colonization in allo-HSCT patients and to examine the impact of oropharyngeal Enterobacteriaceae colonization on transplant outcomes and patient prognosis.

## MATERIALS AND METHODS

### Study design

This study is a prospective, observational cohort investigation conducted at the First Affiliated Hospital of Soochow University. From June 2022 to June 2023, we enrolled 145 patients who underwent allo-HSCT in the stem cell transplantation unit. All patients were treated with antimicrobial prophylaxis when allo-HSCT started. Prior to neutropenia, infection prophylaxis with levofloxacin, acyclovir, and voriconazole (in high-risk patients) was implemented to prevent bacterial, viral, and fungal infections, respectively. During neutropenia, cefotaxime was used for infection prevention. Before initiating the conditioning regimen, we performed oropharyngeal and anal swabs on all patients to culture and screen for colonizing pathogens. Concurrently, we conducted mNGS testing. We excluded 17 patients from the analysis based on the criteria outlined in Fig. 1, resulting in a final cohort of 128 patients. Detailed inclusion and exclusion criteria, as well as the study workflow, are illustrated in Fig. 1. The Ethics Committee of the First Affiliated Hospital of Soochow University reviewed and approved this study. All enrolled patients provided informed consent by signing the consent form prior to sample collection.

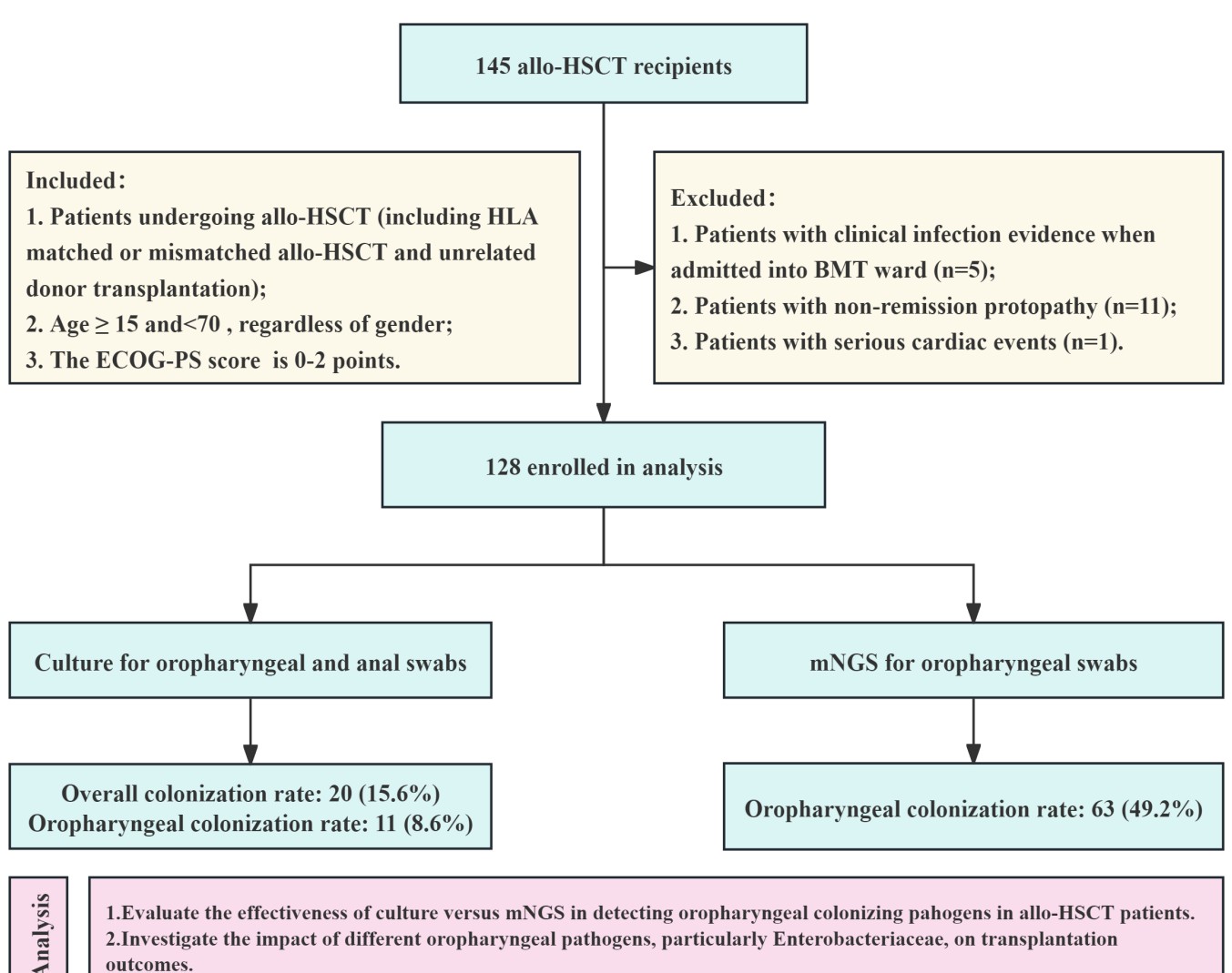

**FIG 1** Flowchart of this study. Detailed inclusion and exclusion criteria are included.

## Sample collection

Upon entering the stem cell transplantation unit, patients followed a strict oral care regimen. Each morning and before bedtime, they used a chlorhexidine acetate mouthwash (containing chlorhexidine acetate, menthol, and glycerin) and brushed their teeth with a sponge stick. Before meals, they rinsed with room-temperature purified water, and immediately after meals, they used either room-temperature purified water or 0.9% saline. Each rinse consisted of 15–30 mL of solution, swished for 30 seconds while performing cheek-puffing motions 10–15 times. Patients adhered closely to dietary guidelines as per transplantation requirements, with particular focus on food quality and disinfection protocols. Patients with dentures removed them twice daily for cleaning with saline. For patients receiving melphalan, we applied cryotherapy using ice cubes or ice water starting 5 minutes before administration and continuing until 30 minutes post-infusion. After low-dose methotrexate treatment, we used calcium leucovorin mouthwash. Before collecting oropharyngeal swabs, patients rinsed their mouths 3–4 times with sterile saline. We obtained the swab by rubbing a sterile swab on the posterior oropharyngeal wall multiple times, then placed it in a sterile tube, sealed, and sent it for analysis. Oropharyngeal swabs were collected before the initiation of conditioning therapy and tested via both culture and mNGS. Throughout the collection process, we adhered strictly to aseptic techniques to prevent sample contamination.

## Testing methods

### Culture

We inoculated oropharyngeal swabs onto Columbia blood agar plates and incubated them at 35°C in an aerobic environment for 48 hours. The normal oropharyngeal flora primarily included *Streptococcus viridis*, *Streptococcus salivarius*, *Veillonella*, and non-pathogenic species of *Neisseria*, excluding *Neisseria gonorrhoeae,* and *Neisseria meningitides* (14, 15). If the presence of Enterobacteriaceae was detected, the corresponding patient was classified as having oropharyngeal colonization with Enterobacteriaceae. We further performed carbapenem resistance testing on these Enterobacteriaceae-positive samples. We inoculated the isolates onto meropenem chromogenic agar plates and incubated them at 35 ± 2°C for 18–48 hours. We identified blue colonies as carbapenem-resistant target bacteria, including *Klebsiella pneumoniae*, *Enterobacter cloacae*, *Escherichia coli*, and *Citrobacter species*. For genotypic analysis, we performed PCR amplification for the KPC, NDM, IMP, VIM, and OXA-48 genes. If we observed discrepancies between genotypic and phenotypic resistance profiles, we amplified full-length resistance genes and subjected them to Sanger sequencing. The patients corresponding to these samples were classified as having oropharyngeal colonization with carbapenem-resistant Enterobacteriaceae (CRE). We then subjected these isolates to mass spectrometry for identification and antimicrobial susceptibility testing.

### mNGS

After eluting the oropharyngeal swab samples, we extracted free DNA using a large-volume nucleic acid extraction kit based on the magnetic bead method. We measured DNA concentration and purity with a NanoDrop 2000 spectrophotometer. We randomly fragmented the DNA and constructed libraries through end-repair, adapter ligation, amplification, and purification, followed by quality control and quantitative pooling. We performed sequencing using the NextSeq 550Dx sequencer. We downloaded pathogen genome sequences from the NCBI RefSeq and GenBank databases, which included bacteria, viruses, fungi, and parasites. We also retrieved the human GRCh37 reference genome sequence from the UCSC database to use as the host reference. After removing human genome sequences to facilitate de-hosting, we aligned the non-host sequences with known sequence databases. To exclude environmental and laboratory contaminants, an external negative control (No Template Control, NTC) was included in each mNGS assay run in our study. Common colonized bacteria and contaminating

microorganisms, including *Rothia*, *Corynebacterium*, *Actinomyces israelii*, *Neisseria,* and *Streptococcus*, were considered microecological bacteria in our study (16).

Antibiotic resistance functional annotation information for the amino acid sequences of the non-redundant gene set was obtained using AMRFinderPlus. The abundance of each antibiotic resistance function was then calculated by summing the gene abundances corresponding to the respective resistance functions.

## Statistical analysis

We created contingency tables for qualitative variables and calculated the median and interquartile range (IQR) for quantitative variables. We analyzed categorical variables using Pearson's chi-squared test and Fisher's exact test. To compare continuous variables, we employed the Mann-Whitney *U* test and Student's *t*-test. We used the cumulative incidence function and competing risk models to examine factors influencing engraftment, post-transplant complications, relapse, and NRM, treating death from other causes as competing events. We calculated *P*-values for these events using the Fine-Gray method. To investigate factors affecting overall survival (OS) and progression-free survival (PFS), we utilized Cox proportional hazards models. In the univariate analysis, we presented results as hazard ratios (HR) with 95% confidence intervals (CI) and *P*-values. We included variables with *P*-values < 0.1 from the univariate analysis in the multivariate Cox regression analysis. We conducted survival analysis using the Kaplan-Meier method. We performed all data analyses using R software (version 4.2.0), and all statistical tests were two-sided, with a significance level set at $P < 0.05$.

## RESULTS

### Clinical characteristics and test results

Based on the inclusion and exclusion criteria outlined in Fig. 1, we included 128 of the 145 patients undergoing allo-HSCT in this study. Table 1 presents the clinical characteristics of these patients. In accordance with the standard transplantation protocol, we conducted oropharyngeal and anal swab cultures for all patients prior to transplantation to evaluate colonization status. Fig. 2A illustrates the oropharyngeal and anal colonization detected in the 128 patients using conventional culture methods. The colonization rate of oropharyngeal pathogenic microorganism was 8.6% (11/128), which comprised 4 cases of *Klebsiella pneumoniae*, 4 cases of *Stenotrophomonas maltophilia*, 2 cases of *Enterobacter cloacae*, and 1 case of *Burkholderia cenocepacia*. The detection rate of anal swab cultures was 10.2% (13/128), including 8 cases of *Escherichia coli*, 4 cases of *Klebsiella pneumoniae*, and 1 case of *Enterobacter cloacae*. By combining the results of the oropharyngeal and anal swab cultures, we determined an overall colonization rate of 15.6% (20/128), with Enterobacteriaceae colonization observed in 11.7% (15/128) of the patients.

Among the 128 patients, only 5 exhibited negative oropharyngeal results mNGS, while 123 demonstrated positive results. mNGS detected viruses in 91.4% (117/128) of the patients and identified bacteria or fungi in 49.2% (63/128). The detection rate of mNGS significantly exceeded that of conventional culture methods (8.6% vs 49.2%, $P <$ 0.001). Fig. 2B summarizes the overall detection of bacteria and fungi in the oropharyngeal mNGS samples from the 128 patients. Given that nearly all patients displayed viral colonization in the upper respiratory tract, we excluded viruses from further analysis, instead focusing on bacterial and fungal colonization and species distribution (Fig. 2C). Among the detected organisms, *Acinetobacter baumannii* was the most frequently identified bacterium via mNGS (17.2%, 22/128) despite its absence in traditional culture methods (Fig. 2A). Fungi were detected from six patients through mNGS testing. Additionally, mNGS revealed a high oropharyngeal colonization rate of Enterobacteriaceae (18.0%, 23/128).

**TABLE 1** Baseline clinical characteristics of 128 patients[a]

| Characteristics | Cases (n = 128) |
|---|---|
| Male gender (%) | 64 (50.0) |
| Age (median [IQR]) | 44.00 (30.00, 53.00) |
| Underlying diseases (%) | |
| AA/PNH/AA-PNH | 20 (15.6) |
| ALL | 15 (11.7) |
| AML | 53 (41.4) |
| Lymphoma | 4 (3.1) |
| MDS | 26 (20.3) |
| MPAL | 5 (3.9) |
| Others | 5 (3.9) |
| HCT-CI (median [IQR]) | 1.00 (1.00, 2.00) |
| Days from diagnosis to transplantation (median [IQR]) | 167.00 (94.95, 342.00) |
| Times of chemotherapy (median [IQR]) | 3.71 (1.75, 5.00) |
| HLA antibody positive (%) | 45 (35.2) |
| Donor (%) | |
| Haploid | 90 (70.3) |
| MMUD | 22 (17.2) |
| MSD | 8 (6.3) |
| MUD | 8 (6.3) |
| ABO compatibility (%) | 64 (50.0) |
| Graft source (%) | |
| PB | 113 (88.3) |
| PB + BM | 2 (1.6) |
| UCB | 13 (10.1) |
| ATG use (%) | 115 (89.8) |
| Cord blood use (%) | 57 (44.5) |
| MNC (10E8/kg), median (IQR) | 14.05 (8.86, 18.53) |
| CD34+(10E6/kg), median (IQR) | 5.79 (3.86, 7.83) |
| CD3+(10E6/kg), median (IQR) | 1.82 (1.45, 2.09) |

[a]AA, aplastic anemia; PNH, paroxysmal nocturnal hemoglobinuria; ALL, acute lymphoblastic leukemia; AML, acute myelogenous leukemia; MDS, myelodysplastic; MPAL, mixed phenotype acute leukemia; HCT-CI, hematopoietic cell transplantation–comorbidity index; MMUD, mismatched unrelated donor; MSD, matched sibling donor; MUD, matched unrelated donor; PB, peripheral blood; BM, bone marrow; UCB, umbilical cord blood; MNC, mononuclear cells.

## Overview of enterobacteriaceae colonization

Based on the results from mNGS and culture analyses of oropharyngeal swabs, we identified 23 patients with Enterobacteriaceae oropharyngeal colonization prior to allo-HSCT. The detected Enterobacteriaceae species included *Klebsiella pneumoniae*, *Enterobacter cloacae*, and *Escherichia coli* (Table S1). Among the six patients with positive oropharyngeal swabs for Enterobacteriaceae by culture, the mNGS method also identified Enterobacteriaceae at the same time point. Of these, five patients exhibited consistent results between mNGS and culture, resulting in a concordance rate of 83.3%. One patient's culture identified *Klebsiella pneumoniae*, while mNGS detected *Enterobacter cloacae*; notably, this patient did not experience any infection events during the subsequent transplantation process. An additional 17 patients had negative oropharyngeal culture results but tested positive for Enterobacteriaceae via mNGS. Among the 23 patients with Enterobacteriaceae detected by mNGS, 12 were positive for *Klebsiella pneumoniae*, 7 for *Enterobacter cloacae*, 2 exhibited both *Klebsiella pneumoniae* and *Enterobacter cloacae*, and 2 were positive for *Escherichia coli*.

Table S2 compares the clinical characteristics of patients with and without Enterobacteriaceae colonization. Patients in the Enterobacteriaceae colonization group were significantly older (P = 0.002). Additionally, we observed a statistically significant difference in underlying diseases distribution between the two groups (P = 0.036); 15

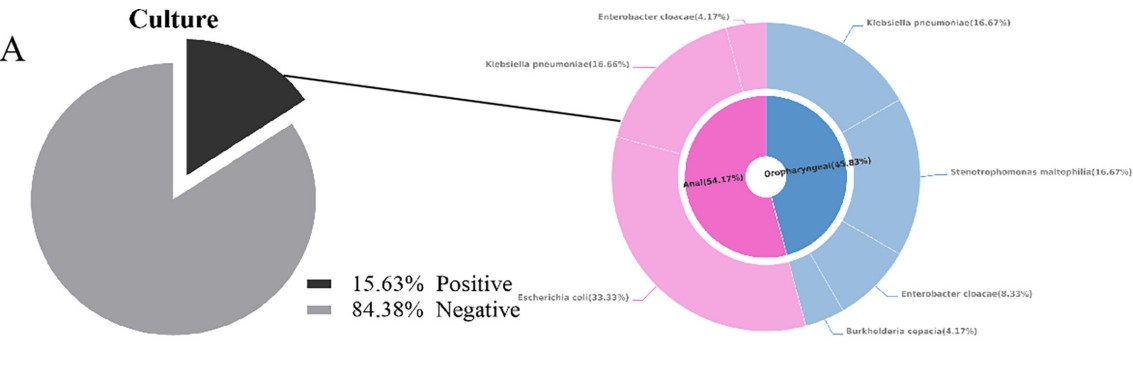

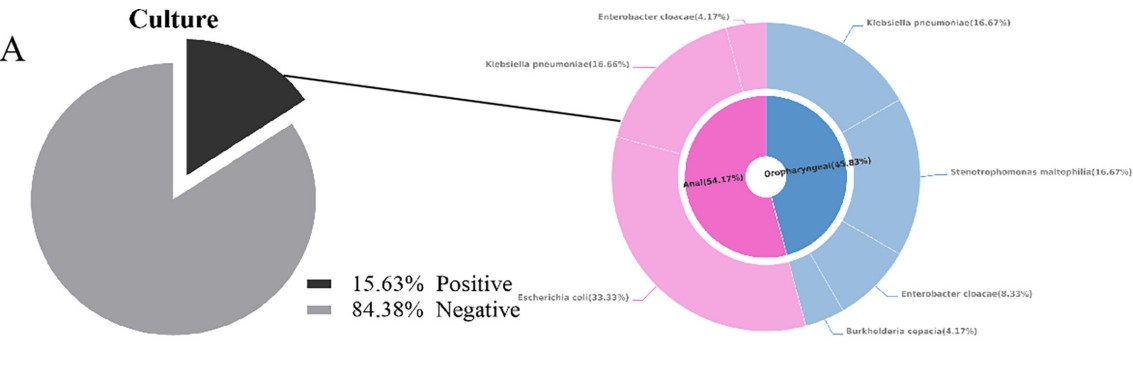

FIG 2  Overview of culture and mNGS detection. (A) Culture of the oropharyngeal swabs and anal swabs. (B) mNGS of the oropharyngeal swabs. (C) Distribution of the bacterial and fungal strains as detected by mNGS.

patients (65.2%) in the Enterobacteriaceae colonization group had acute myeloid leukemia (AML), compared to 36 patients (34.3%) in the non-colonization group. Notably, none of the 26 patients with myelodysplastic syndromes (MDS) exhibited oropharyngeal Enterobacteriaceae colonization. No significant differences appeared in other clinical characteristics between the two groups.

## Clinical significance of colonization screening based on culture methods

To investigate how different screening methods for oropharyngeal colonization affect transplant complications and outcomes, we first assessed the clinical significance of pathogen colonization identified through culture techniques. Our results revealed that the incidence of Epstein-Barr virus (EBV) infection post-transplantation was significantly higher in the colonization group than in the non-colonization group (30% vs 11.1%, $P = 0.016$). However, we observed no significant differences between the two groups regarding neutrophil and platelet engraftment, cytomegalovirus (CMV) infection, post-transplant BSI, acute graft-versus-host disease (aGVHD), hemorrhagic cystitis, veno-occlusive disease (VOD), thrombotic microangiopathy (TMA), relapse, NRM, OS, or PFS (Fig. S1). Univariate and multivariate analyses, adjusted for baseline and transplant-related characteristics, indicated that colonization identified through culture methods serves as an independent risk factor for post-transplant EBV infection (HR 4.38; 95% CI: 1.46–13.20; $P = 0.009$) (Table S3).

## Clinical significance of oropharyngeal colonization screening by mNGS

We further assessed the impact of oropharyngeal colonization identified through mNGS on transplant outcomes. Our results indicated that the incidence of VOD or TMA was significantly higher in patients with bacterial or fungal colonization than in those without (14.3% vs 6.2%, $P = 0.012$). However, we found no significant differences between the two groups regarding neutrophil and platelet engraftment, CMV infection, EBV infection, post-transplant BSI, aGVHD, hemorrhagic cystitis, NRM, OS, or PFS (Fig. S2). Multivariate analysis revealed that colonization detected by mNGS serves as an independent risk factor for post-transplant VOD or TMA (HR 3.84; 95% CI: 1.49–9.89; $P = 0.005$) (Table S4). Although the detection rate of oropharyngeal fungal colonization is low, results revealed that one patient, whose oropharyngeal swab tested positive for *Pneumocystis jirovecii*, developed Pneumocystis jirovecii pneumonia (PJP) after allo-HSCT and subsequently died.

When we evaluated the impact of Enterobacteriaceae colonization detected by mNGS on transplant complications and outcomes, we found that patients with oropharyngeal Enterobacteriaceae colonization had a higher incidence of post-transplant BSI (39.1% vs 19.0%, $P = 0.033$)(Fig. 3C). This colonization emerged as an independent risk factor for BSI (HR 2.34; 95% CI: 1.08–5.10; $P = 0.032$) (Table 2). The incidence of VOD or TMA was also greater in the Enterobacteriaceae colonization group compared to the non-colonization group (17.4% vs 8.5%, $P = 0.04$).(Fig. 3B) However, multivariate analysis revealed that Enterobacteriaceae colonization was not an independent risk factor for VOD or TMA (HR 2.66; 95% CI: 0.85–8.31; $P = 0.092$) (Table S5). Notably, we observed a trend toward delayed platelet engraftment in patients with pre-transplant Enterobacteriaceae colonization although this finding did not reach statistical significance ($P = 0.08$)(Fig. 3A). Furthermore, we found no significant differences in neutrophil engraftment, CMV infection, EBV infection, aGVHD, or hemorrhagic cystitis between the two groups.

Cumulative incidence function (CIF) and Kaplan-Meier (KM) curves revealed that patients with Enterobacteriaceae colonization experienced significantly poorer transplant outcomes. Although relapse rates did not differ significantly between the groups ($P = 0.5$), patients with oropharyngeal Enterobacteriaceae colonization exhibited worse OS, higher NRM, and poorer PFS compared to those without colonization ($P$-values: 0.009, 0.0037, and 0.0046, respectively)(Fig. 3D-F). Further multivariate analysis confirmed that oropharyngeal Enterobacteriaceae colonization is an independent risk factor for NRM

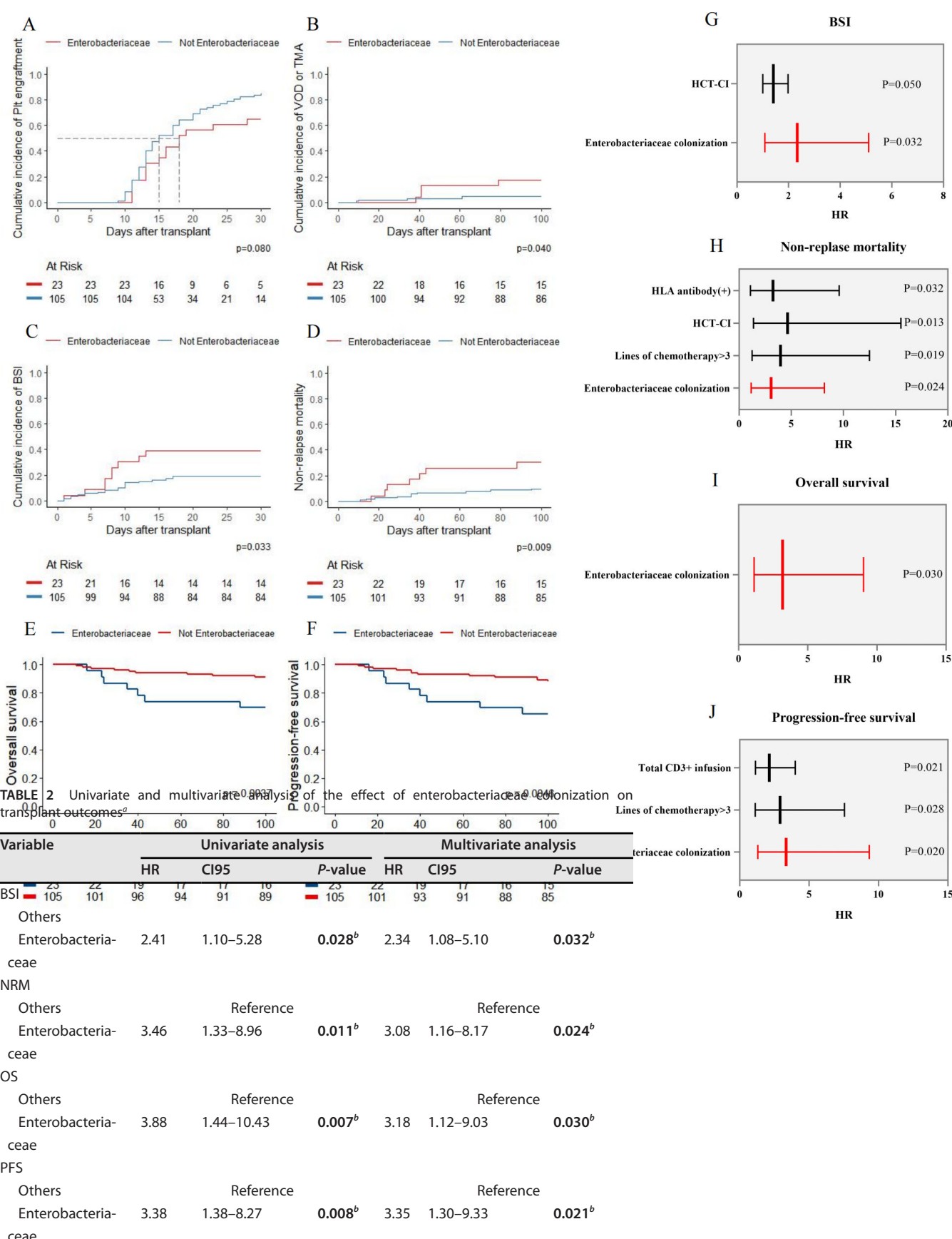

**TABLE 2** Univariate and multivariate analysis of the effect of enterobacteriaceae colonization on transplant outcomes[a]

| Variable | Univariate analysis | | | Multivariate analysis | | |
|---|---|---|---|---|---|---|
| | HR | CI95 | P-value | HR | CI95 | P-value |
| BSI | | | | | | |
| Others | | Reference | | | Reference | |
| Enterobacteria-ceae | 2.41 | 1.10–5.28 | **0.028**[b] | 2.34 | 1.08–5.10 | **0.032**[b] |
| NRM | | | | | | |
| Others | | Reference | | | Reference | |
| Enterobacteria-ceae | 3.46 | 1.33–8.96 | **0.011**[b] | 3.08 | 1.16–8.17 | **0.024**[b] |
| OS | | | | | | |
| Others | | Reference | | | Reference | |
| Enterobacteria-ceae | 3.88 | 1.44–10.43 | **0.007**[b] | 3.18 | 1.12–9.03 | **0.030**[b] |
| PFS | | | | | | |
| Others | | Reference | | | Reference | |
| Enterobacteria-ceae | 3.38 | 1.38–8.27 | **0.008**[b] | 3.35 | 1.30–9.33 | **0.021**[b] |

[a]BSI, bloodstream infection; NRM, non-relapse mortality; OS, overall survival; PFS, progression-free survival.
[b]P value less than 0.05.

Fig 3 (Continued)

**FIG 3** (A–E) Comparison of post-transplant complications and prognosis between patients with and without Enterobacteriaceae as detected by a pre-transplant oropharyngeal mNGS test, including (A) platelet engraftment, (B) VOD or TMA, (C) BSI, (D) NRM, (E) OS, and (F) PFS. (G–J) Multivariate analysis for BSI (G), NRM (H), OS (I), and risk of PFS (J).

(HR 3.08; 95% CI: 1.16–8.17; *P* = 0.024), OS (HR 3.18; 95% CI: 1.12–9.03; *P* = 0.030), and PFS (HR 3.35; 95% CI: 1.30–9.33; *P* = 0.021)(Fig. 3G-J) (Table 2).

## Impact of oropharyngeal CRE colonization on transplant outcomes

We identified five patients with oropharyngeal colonization by CRE. Three of these were *Enterobacter cloacae*, with the NDM-1 resistance gene detected through mNGS. The other two were *Klebsiella pneumoniae*, which did not show resistance genes but were confirmed as CRKP through antimicrobial susceptibility testing. Table S6 presents the detection of resistance genes in the five CRE samples, along with infection outcomes and post-transplant results.

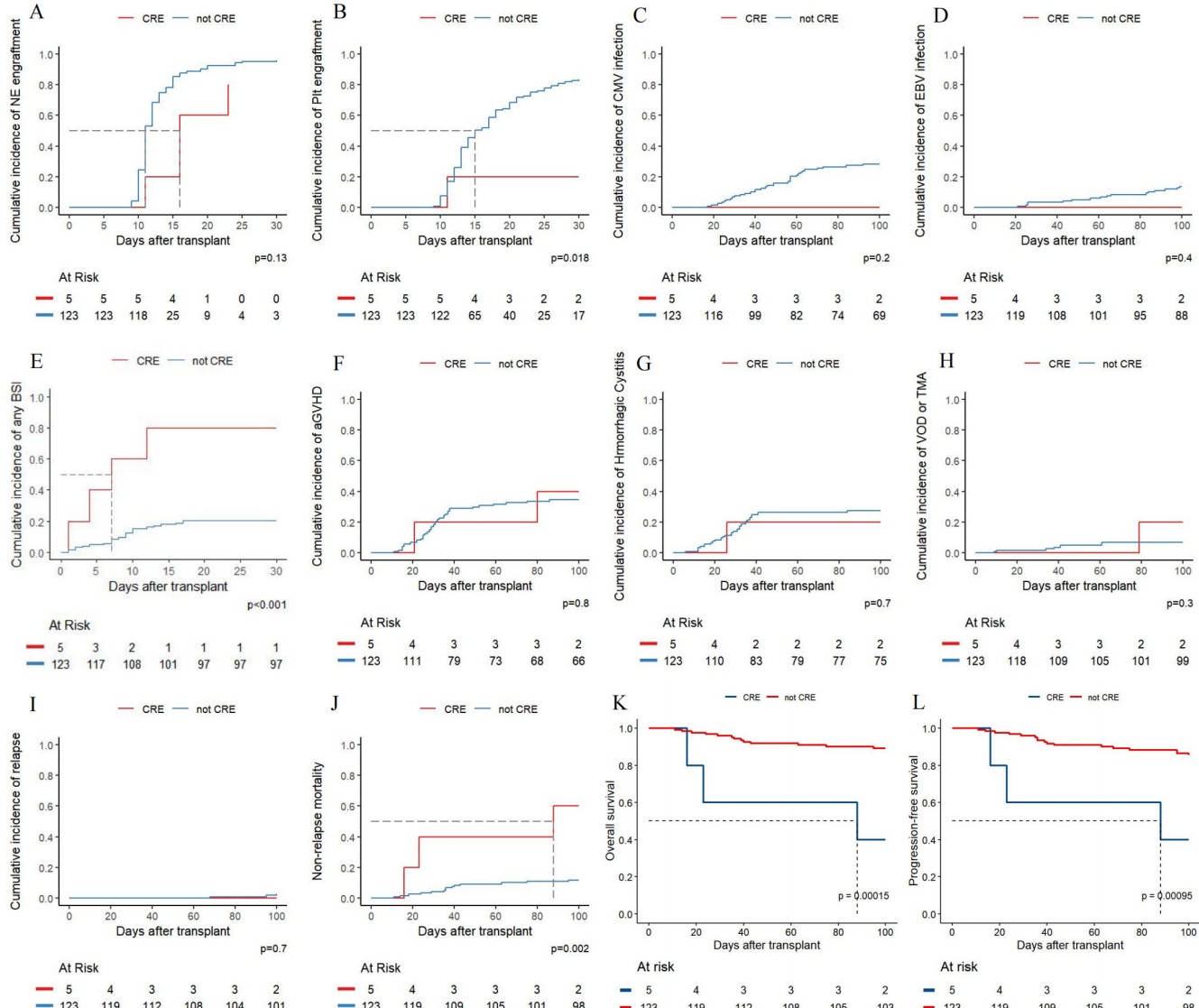

**FIG 4** Comparison of post-transplant complications and prognoses between patients with and without CRE colonization, including (A) neutrophil engraftment, (B) platelet engraftment, (C) CMV infection, (D) EBV infection, (E) BSI, (F) aGVHD, (G) hemorrhagic cystitis, (H) VOD or TMA, (I) relapse, (J) NRM, (K) OS, and (L) PFS.

Although only five of the 128 patients in our study had oropharyngeal CRE colonization (2 cases of CRKP and 3 cases of carbapenem-resistant *Enterobacter cloacae*), we compared transplant outcomes between those with and without CRE colonization. As shown in Fig. 4, patients with oropharyngeal CRE colonization experienced delayed platelet engraftment ($P = 0.018$) and a significantly higher incidence of post-transplant BSI ($P < 0.001$). Additionally, these patients exhibited worse OS ($P = 0.002$), higher non-relapse mortality (NRM) ($P = 0.00015$), and poorer PFS ($P = 0.00095$).

## DISCUSSION

In this prospective study, we compared the performance of mNGS with conventional culture methods in detecting oropharyngeal colonization in allo-HSCT patients. While mNGS improved the detection rate of oropharyngeal colonization, not all positive results were clinically relevant. Colonization by Enterobacteriaceae or CRE was identified as a key factor contributing to post-transplant infections and poor outcomes.

With the rapid advancements in mNGS, it has become widely used in microbial genomics, pathogen identification, and antibiotic resistance monitoring (17, 18). A study of 442 febrile neutropenia patients with acute leukemia showed that mNGS significantly outperformed blood cultures in both diagnostic accuracy and speed. Furthermore, mNGS was less affected by prior antimicrobial use, making it a valuable tool for diagnosing infections in febrile neutropenia patients unresponsive to empirical antibiotic therapy (19). However, the role of mNGS in screening colonizing pathogens remains unclear. Routine pre-transplant screening typically involves oropharyngeal and anal swabs. A study by A. Forcina found that rectal colonization with multidrug-resistant Gram-negative bacteria (MDR-GNB) before HSCT did not significantly impact OS, transplant-related mortality (TRM), or infection-related mortality (IRM) (20). Similarly, a study on vancomycin-resistant enterococci (VRE) demonstrated that intestinal VRE colonization prior to transplantation did not increase TRM (21). Based on these findings, we focused on oropharyngeal colonization and conducted a prospective observational study of 128 allo-HSCT patients to evaluate oropharyngeal colonization using mNGS.

Previous virome data indicate that healthy individuals can test positive for various viruses through mNGS (22), whereas cultures typically detect only bacteria and fungi. Therefore, we excluded viral results from our mNGS data in this study. Earlier studies using culture methods reported oropharyngeal colonization detection rates ranging from 10.0% to 20.0% (7, 10). When excluding the detection of non-pathogenic organism, our findings show that mNGS significantly improves the detection rate of oropharyngeal colonization, with Enterobacteriaceae being the most commonly identified organism by both mNGS and culture methods. In healthy individuals, Enterobacteriaceae species such as *Escherichia coli* and *Klebsiella* are part of the normal gut microbiota, colonizing the gastrointestinal tract (23). However, in immunocompromised or long-term hospitalized patients, Enterobacteriaceae can also colonize the upper respiratory tract (24). In our study, *Klebsiella pneumoniae* was the most frequently detected oropharyngeal colonizer. Our previous research demonstrated that pre-transplant colonization with *Klebsiella pneumoniae* negatively impacts transplant outcomes (25).

Infection is a leading cause of early mortality following HSCT. Patients who develop BSIs post-transplant experience higher NRM and poorer OS (26). In our study, we found no significant correlation between colonization—identified by culture—and an increased risk of BSI after transplantation, consistent with previous studies (10). However, we noted a higher incidence of EBV infection in patients with colonization detected by culture. This may result from microbial colonization, which can lead to dysbiosis, chronic inflammation due to colonizing pathogens, and impaired T-cell function (27). Additionally, we observed an increased risk of VOD and TMA in patients with oropharyngeal colonization identified through mNGS after allo-HSCT. Both VOD and TMA are linked to endothelial damage (28, 29). Microbial colonization may disrupt the oropharyngeal microbiota, further compromising the mucosal barrier and promoting the release of pro-inflammatory cytokines, such as TNF-α and IL-6, which contribute to endothelial

injury (30). Despite these observations, we did not find that colonization adversely affected the prognosis of allo-HSCT.

Our investigation into the effects of oropharyngeal Enterobacteriaceae colonization on transplant outcomes revealed that such colonization increases the risk of BSI and VOD or TMA and negatively impacts transplant prognosis. Colonization by CRE had particularly severe consequences. Among the five patients colonized with CRE, four developed BSIs from the same strain during transplantation, and three died early in the post-transplant period due to severe infections. These findings highlight the critical need for heightened attention to oropharyngeal CRE colonization in clinical practice. In recent years, factors such as inappropriate antibiotic use, horizontal gene transfer, genetic mutations, selective pressure, and hospital transmission have contributed to the rapid emergence of resistance genes. This has made Enterobacteriaceae resistance a significant global public health issue (31). Resistance genes like NDM, KPC, and MCR have been increasingly reported in recent years (32–35). Previous studies also demonstrated that perianal CRE colonization worsens transplant outcomes (36). Thus, early identification of patients colonized with CRE is essential for preventing infections and limiting the spread of resistance genes (37).

We have confirmed the harmful impact of Enterobacteriaceae colonization in the oropharynx on allo-HSCT outcomes. So, is there any intervention that could help mitigate this risk? In our research, all allo-HSCT patients received antibiotic prophylaxis to prevent infections. Decolonization is not typically a routine practice for patients with CRE colonization (38). However, our study has shown that oropharyngeal CRE colonization is a significant risk factor for subsequent infections and prognosis. Therefore, decolonization can be considered necessary for immunocompromised populations like those undergoing allo-HSCT. However, early detection of CRE colonization through screening testing proved to be a more important tool to control CRE spread and subsequent infection (39).

This prospective study is the first to examine the impact of oropharyngeal Enterobacteriaceae colonization on outcomes following allo-HSCT while also evaluating the utility of mNGS for detecting colonization. However, the study's limitations include a small sample size and the absence of a detailed analysis of the antibiotic resistance profiles of oropharyngeal Enterobacteriaceae. Future studies with larger cohorts are needed to validate these findings. Additionally, using mNGS to better understand the resistance mechanisms of colonizing pathogens could offer valuable insights for clinical practice.

## Conclusion

In conclusion, our findings demonstrate that mNGS holds significant value for screening oropharyngeal colonization prior to allo-HSCT. Clinicians should interpret mNGS results cautiously, as not all colonization has clinical relevance. However, oropharyngeal colonization by Enterobacteriaceae, CRE, warrants serious attention, and in certain cases, decolonization strategies may be necessary.

### ACKNOWLEDGMENTS

We thank the participants in the study.

This work was supported by the National Natural Science Foundation of China (Grant Nos. 82170222); the Jiangsu Natural Science Foundation (BK20211070); The Key Disease Program of Suzhou (LCZX202101); Priority Academic Program Development of Jiangsu Higher Education Institutions (PAPD); Research project of Jiangsu Provincial Health Commission (ZD2021008).

X.W. offered conceptualization and performed the research, Y.Z. and J.G. analyzed the data, Y.Z. and Yuhang Wang wrote the paper, Z.T. offered methodology, X.M. and X.W. provided clinical samples, X.W., J.L., and D.W. provided funding acquisition.

Y.Z., J.G., Y.W., Z.T., X.M., J.L., D.W., X.W. All authors declare that they have no conflict of interest.

# AUTHOR AFFILIATIONS

[1]First Affiliated Hospital of Soochow University, Suzhou, China

[2]National Clinical Research Center for Hematologic Diseases, Jiangsu Institute of Hematology, Suzhou, Jiangsu, China

[3]Institute of Blood and Marrow Transplantation, Collaborative Innovation Center of Hematology, Soochow University, Suzhou, Jiangsu, China

[4]Department of Otolaryngology Head and Neck Surgery, The First Affiliated Hospital of Soochow University, Suzhou, China

[5]Department of Epidemiology and Statistics, School of Public Health, Faculty of Medicine, Soochow University, Suzhou, Jiangsu, China

# AUTHOR ORCIDs

Yuqi Zhang http://orcid.org/0009-0001-7548-0454
Xiaojin Wu http://orcid.org/0000-0003-3894-0631

# FUNDING

| Funder | Grant(s) | Author(s) |
| --- | --- | --- |
| National Natural Science Foundation of China | 82170222 | Xiaojin Wu |
| Natural Science Foundation of Jiangsu Province | BK20211070 | Xiaojin Wu |
| Priority Academic Program Development of Jiangsu Higher Education Institutions | | Depei Wu |

# AUTHOR CONTRIBUTIONS

Yuqi Zhang, Formal analysis, Writing – original draft | Jianrong Ge, Data curation, Resources | Yuhang Wang, Writing – original draft | Zaixiang Tang, Methodology | Xiao Ma, Resources | JIsheng Liu, Funding acquisition, Supervision | Depei Wu, Funding acquisition, Supervision | Xiaojin Wu, Conceptualization, Funding acquisition, Supervision

# ETHICS APPROVAL

The study was conducted in accordance with the principles of the Decla ration of Helsinki and approved by the ethics committee of the First Affiliated Hospital of Soochow University (registry number 213/2022). Informed consents were obtained before mNGS.

# ADDITIONAL FILES

The following material is available online.

## Supplemental Material

**Supplemental material (Spectrum00028-25-S0001.docx).** Fig. S1 and S2; Tables S1 to S6.

## Open Peer Review

**PEER REVIEW HISTORY (review-history.pdf).** An accounting of the reviewer comments and feedback.

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
