## [Reviewer comments · Microbiology Spectrum]

Microbiology Spectrum

Clinical Value of Metagenomic Next-Generation Sequencing in Screening Oropharyngeal Colonization in Patients Undergoing Allogeneic Hematopoietic Stem Cell Transplantation: A Prospective Observational Study

Yuqi Zhang, Jianrong Ge, Yuhang Wang, Zaixiang Tang, Xiao Ma, JIsheng Liu, Depei Wu, and Xiaojin Wu

Corresponding Author(s): Xiaojin Wu, First Affiliated Hospital of Soochow University

Review Timeline:

Submission Date:	January 6, 2025
Editorial Decision:	February 7, 2025
Revision Received:	March 26, 2025
Accepted:	April 9, 2025

Editor: Po-Yu Liu

Reviewer(s): Disclosure of reviewer identity is with reference to reviewer comments included in decision letter(s). The following individuals involved in review of your submission have agreed to reveal their identity: Brian Scott Grundy (Reviewer #2)

Transaction Report:

DOI: <https://doi.org/10.1128/spectrum.00028-25>

Re: Spectrum00028-25 (Clinical Value of Metagenomic Next-Generation Sequencing in Screening Oropharyngeal Colonization in Patients Undergoing Allogeneic Hematopoietic Stem Cell Transplantation: A Prospective Observational Study)

Dear Prof. Xiaojin Wu:

Thank you for the privilege of reviewing your work. Below you will find my comments, instructions from the Spectrum editorial office, and the reviewer comments.

Revision Guidelines

Sincerely,
Po-Yu Liu
Editor
Microbiology Spectrum

Reviewer #1 (Comments for the Author):

An interesting paper focusing on oropharyngeal colonization and the impact on patients receiving allo-HSCT.

A few things that might benefit from clarification:

- It's not clear why *Enterobacter* colonies (from culture) were only inoculated onto meropenem chromogenic agar. Was this just

to evaluate resistance to meropenem on samples that would otherwise be considered culture positive for Enterobacter or are samples that were successfully cultured on the meropenem chromogenic agar the only samples classified as being colonized? If it's the latter, and only patients with cultures growing on meropenem were considered "colonized", I wonder if this might be missing out on positive cultures for meropenem-susceptible Enterobacter, and possibly explain some of the differences between culture and mNGS results. The mNGS results would account for all Enterobacter in the sample, not just CRE, so comparing all Enterobacter sequence data to just CRE culture data could possibly explain some of the differences.

- Were any negative control swabs processed alongside patient swabs? Or were there any other processing controls to make sure that there weren't reagent contaminants that could potentially mediate some of the differences in culture vs mNGS results?

- Were there any alignment thresholds/coverage levels identified for classifying a mNGS sample as "colonized"? A quick explanation of which criteria were used to identify a sample as colonized vs a sample that might have had a small amount of spurious or misclassified reads but did not actually contain the bacteria would be helpful.

Reviewer #2 (Comments for the Author):

The Authors present an interesting manuscript exploring the utility of mNGS for screening of pathogens that may result in worse outcomes in patients undergoing allogenic stem cell transplants. They find that mNGS of oropharyngeal swabs is more sensitive than routine cultures at detecting pathogenic organisms that can result in worse outcomes. They find that those with enterobacterales or Enterobacteriaceae (needs to be corrected from Enterobacter) colonization have increase adverse outcomes associated with SCT.

This manuscript shows some novel insights and possible benefit of using mNGS for screening prior to transplant but it should further elaborate on how routine screening methods have not shown any benefit and that mNGS may have benefit in identifying at risk patients and suggesting possible interventions or further randomized studies to improve outcomes.

Throughout the paper, "Enterobacter" needs to be corrected to correctly identify the order of enterobacterales or family Enterobacteriaceae. Currently the manuscript uses Enterobacter to describe a group of organisms incorrectly, when Enterobacter only refers to the genus.

Overall, the methods and analysis in this study are sound but can use some clarity in how pathogens were identified or excluded from analysis. For example, in oropharyngeal and rectal swabs there should always be microorganisms that would be detected as normal flora and are often routinely excluded in the microbiology lab in clinical reporting. Similarly, mNGS in a non sterile space like the oropharynx should have significantly more pathogens detected than reported in this manuscript, and the manuscript would benefit from how these non-pathogens were excluded. Additionally, there should have been significantly more enterobacterales identified in rectal swabs (unless this is only detecting CRE). The inclusion of rectal swab data adds little to this manuscript other than to show routine screening is not very effective, but does not relate well to oropharyngeal mNGS. Additionally figure 2C is difficult to read which column matches with which organism. The Fungi identified by mNGS are interesting and would be good to comment on them even though small number. Did this patient have poor outcomes or any fungal infections. Phenotypic data is described in the methods but not described elsewhere, would be interesting if any analysis if any resistance identified associated with outcomes otherwise would consider removing.

Figure 3 and 4 is a lot of data to review and not addressed well in the manuscript. It should either be simplified or explained further in the manuscript. Additionally, I am not sure kaplan-meier curves add much to the overall hazard ratios described in the manuscript.

Specific Comments:

Abstract:

29-30: I do not consider colonization screening as routine care. Is this screening for MRSA, CRE colonization? Any studies to support this or the benefits to justify screening?

30-32: I do not believe mNGS has been widely adopted in clinical practice. It has a small niche use in rare, atypical or uncultivable infections or in those already receiving antibiotics where culture growth is unlikely.

38-39: This is not unexpected but would specify that organisms were detected not colonization rate. As every individual is colonized with organisms in their oropharyngeal space as it is a non-sterile site. There colonization rate is 100% but you are changing the rate of detection based on diagnostic modality.

Introduction:

73-79: would elaborate on this study further. Would consider adding what the rate of pathogen culture detection is in a normal host oropharyngeal culture for comparison. The detection rate does not highlight the clinical importance of screening. I would include any differences in outcomes based on screening detection which would highlight clinical importance.

77: Klebisella and Enterobacter are Enterobacteriaceae species, not Enterobacter species

80: I don't think there are any clinically relevant oropharyngeal organisms that take weeks to culture.

85-87: yes mNGS is being used in several clinical sequencing tasks, but is still not clinically superior to cultures for pathogen detection in most cases.

91-93: again mNGS is not widely used in routine clinical medicine but is occasionally used as an adjunct to routine culture, antibody, antigen, and/or PCR detection.

94-97: I agree with these limitations but would also include significantly increased costs compared to routine methods, and specialized trained personnel, and often turnaround time is still equal if not longer than routine detection methods.

Methods:

116: not sure how much of this routine neutropenic hygiene protocol is necessary. Would consider adding if patients were on specific antibiotic/antifungal prophylaxis for neutropenia.

139: would specify streptococcus species. While many are normal flora, several species are pathogen and even normal flora strep species can lead to bacteremia especially in neutropenic patients with mucositis.

159-167: this phenotypic testing should be included in the culture methods, including PCR sequencing as this is part of routine culture methods and not mNGS. Then would add how any resistance genes may have been detected using mNGS.

Results:

194: clarify colonization rate as colonization with pathogenic organisms.

197-198: Please clarify the rectal colonization rate. As all of these organisms can be considered normal flora of rectal swabs and I would suspect a much higher rate of colonization with at least one of these organisms in stool samples if not 100%. Are all of these CRE organisms? If so should specify so and would be more clinically relevant.

204: does the bacterial mNGS detection rate include only clinically relevant bacterial detections? Were many species excluded or removed. Figure 2C makes me think that only certain species were included in the mNGS results. For example mNGS does not list any strep species but I'm sure some strep was detected by mNGS swabs but excluded just like it was in the culture results?

214: please specify are you using Enterobacter to describe only the genus Enterobacter species (which looks like maybe on 11 detections) or all Enterobacteriaceae family (including E.coli, klebsiella, etc) which would get to 23 detections and what I think you are describing. This needs to be corrected throughout paper as Enterobacter (genus) is not the proper name for this family of bacteria and is misleading and confusing.

232: is disease type referring to cancer type? Please clarify

242: would clarify the definition of EBV infection. Is this detection of EBV virus by PCR? Did EBV infection result in worse outcomes? EBV usually associated with increased mortality risk. Would also include culture colonization and outcome measure HR for OS, NRM, PFS. If no difference with culture colonization please state so.

288: would clarify utility of CRE detection by mNGS. Hard to understand the numbers here, but I think it sounds like mNGS only detected one CRE. The rest were identified by routine methods? Was any CRE detected by rectal swab, which is the routine method for detecting CRE colonization.

Discussion:

337: would clarify that you are likely excluding non-pathogenic organism detection.

359: would clarify by detection using mNGS

Discussion would benefit from further exploration into potential pathophysiology of link between colonization and thrombosis and other clinical examples showing this link and also if this thrombosis lead to worse outcomes. Further analysis into outcomes would be beneficial as well. Colonization and delayed platelet engraftment does not have an obvious pathophysiology explanation and without statistical significance would consider removing. Additionally, I think some exploration and/or suggestions into potential interventions. Ie if colonization was detected or CRE could changes in antibiotic prophylaxis, empiric antibiotics, delay in transplant and/or efforts to decolonize as efforts to improve outcomes. Similarly, some basic data on if these patients are placed on routine antibiotic prophylaxis or what empiric treatment for neutropenic fever looks like.

Comments and Suggestions for the Author:

An interesting paper focusing on oropharyngeal colonization and the impact on patients receiving allo-HSCT.

A few things that might benefit from clarification:

- It's not clear why *Enterobacter* colonies (from culture) were only inoculated onto meropenem chromogenic agar. Was this just to evaluate resistance to meropenem on samples that would otherwise still be considered culture positive for *Enterobacter* or are samples that were successfully cultured on the meropenem chromogenic agar the only samples classified as being colonized? If it's the latter, and only patients with cultures growing on meropenem were considered "colonized", I wonder if this might be missing out on positive cultures for meropenem-susceptible *Enterobacter* and possibly explain some of the differences between culture and mNGS results. The mNGS results would account for all *Enterobacter* in the sample, not just CRE, so comparing all *Enterobacter* sequence data to just CRE culture data could possibly explain some of the differences.
- Were any negative control swabs processed alongside patient swabs? Or were there any other processing controls in place to make sure that there weren't reagent contaminants that could potentially mediate some of the differences in culture vs mNGS results?
- Were there any alignment thresholds/coverage levels identified for classifying a mNGS sample as "colonized"? A quick explanation of which criteria were used to identify a sequenced sample as colonized vs a sample that might have had a small amount of spurious or misclassified reads but did not actually contain the bacteria would be helpful.

Confidential remarks for the Editors

I believe that this paper, in its current form, has data-informed conclusions and it was easy to comprehend. I see no reason why it shouldn't be published based on the data and interpretations themselves as they are currently presented. However, I think that it would benefit from some clarification on the points I mention in the comments/suggestions for the author.

My one main sticking point - which I think can be clarified and doesn't necessarily prevent potential publication, but is something that would need to be addressed - is that if samples with culture positive results for only CRE were scored as colonized, then we're missing out on Enterobacter that are susceptible to meropenem (if any), and it could explain some of the differences between culture and mNGS results. However, if samples are considered positive for Enterobacter from the initial inoculation on the Columbia agar (when present), and then carbapenem resistance is evaluated afterwards as a screen, then this isn't an issue. Similarly, if all samples that were inoculated on the meropenem plates were also resistant to the meropenem, then this wouldn't matter, as it accounts for all isolates of Enterobacter. It's just not clear how Enterobacter isolates are scored with this additional inoculation step, so clarifying this would be helpful to understand the significance of the culture vs mNGS results.

The Authors present an interesting manuscript exploring the utility of mNGS for screening of pathogens that may result in worse outcomes in patients undergoing allogenic stem cell transplants. They find that mNGS of oropharyngeal swabs is more sensitive than routine cultures at detecting pathogenic organisms that can result in worse outcomes. They find that those with enterobacterales or Enterobacteriaceae (needs to be corrected from Enterobacter) colonization have increase adverse outcomes associated with SCT.

This manuscript shows some novel insights and possible benefit of using mNGS for screening prior to transplant but it should further elaborate on how routine screening methods have not shown any benefit and that mNGS may have benefit in identifying at risk patients and suggesting possible interventions or further randomized studies to improve outcomes.

Throughout the paper, "Enterobacter" needs to be corrected to correctly identify the order of enterobacterales or family Enterobacteriaceae. Currently the manuscript uses Enterobacter to describe a group of organisms incorrectly, when Enterobacter only refers to the genus.

Overall, the methods and analysis in this study are sound but can use some clarity in how pathogens were identified or excluded from analysis. For example, in oropharyngeal and rectal swabs there should always be microorganisms that would be detected as normal flora and are often routinely excluded in the microbiology lab in clinical reporting. Similarly, mNGS in a non sterile space like the oropharynx should have significantly more pathogens detected than reported in this manuscript, and the manuscript would benefit from how these non-pathogens were excluded. Additionally, there should have been significantly more enterobacterales identified in rectal swabs (unless this is only detecting CRE). The inclusion of rectal swab data adds little to this manuscript other than to show routine screening is not very effective, but does not relate well to oropharyngeal mNGS. Additionally figure 2C is difficult to read which column matches with which organism. The Fungi identified by mNGS are interesting and would be good to comment on them even though small number. Did this patient have poor outcomes or any fungal infections. Phenotypic data is described in the methods but not described elsewhere, would be interesting if any analysis if any resistance identified associated with outcomes otherwise would consider removing.

Figure 3 and 4 is a lot of data to review and not addressed well in the manuscript. It should either be simplified or explained further in the manuscript. Additionally, I am not sure kaplan-meier curves add much to the overall hazard ratios described in the manuscript.

Specific Comments:

Abstract:

29-30: I do not consider colonization screening as routine care. Is this screening for MRSA, CRE colonization? Any studies to support this or the benefits to justify screening?

30-32: I do not believe mNGS has been widely adopted in clinical practice. It has a small niche use in rare, atypical or uncultivable infections or in those already receiving antibiotics where culture growth is unlikely.

38-39: This is not unexpected but would specify that organisms were detected not colonization rate. As every individual is colonized with organisms in their oropharyngeal space as it is a non-sterile site. There colonization rate is 100% but you are changing the rate of detection based on diagnostic modality.

Introduction:

73-79: would elaborate on this study further. Would consider adding what the rate of pathogen culture detection is in a normal host oropharyngeal culture for comparison. The detection rate does not highlight the clinical importance of screening. I would include any differences in outcomes based on screening detection which would highlight clinical importance.

77: Klebisella and Enterobacter are Enterobacteriaceae species, not Enterobacter species

80: I don't think there are any clinically relevant oropharyngeal organisms that take weeks to culture.

85-87: yes mNGS is being used in several clinical sequencing tasks, but is still not clinically superior to cultures for pathogen detection in most cases.

91-93: again mNGS is not widely used in routine clinical medicine but is occasionally used as an adjunct to routine culture, antibody, antigen, and/or PCR detection.

94-97: I agree with these limitations but would also include significantly increased costs compared to routine methods, and specialized trained personnel, and often turnaround time is still equal if not longer than routine detection methods.

Methods:

116: not sure how much of this routine neutropenic hygiene protocol is necessary. Would consider adding if patients were on specific antibiotic/antifungal prophylaxis for neutropenia.

139: would specify streptococcus species. While many are normal flora, several species are pathogen and even normal flora strep species can lead to bacteremia especially in neutropenic patients with mucositis.

159-167: this phenotypic testing should be included in the culture methods, including PCR sequencing as this is part of routine culture methods and not mNGS. Then would add how any resistance genes may have been detected using mNGS.

Results:

194: clarify colonization rate as colonization with pathogenic organisms.

197-198: Please clarify the rectal colonization rate. As all of these organisms can be considered normal flora of rectal swabs and I would suspect a much higher rate of colonization with at least one of these organisms in stool samples if not 100%. Are all of these CRE organisms? If so should specify so and would be more clinically relevant.

204: does the bacterial mNGS detection rate include only clinically relevant bacterial detections? Were many species excluded or removed. Figure 2C makes me think that only certain species were included in the mNGS results. For example mNGS does not list any strep species but I'm sure some strep was detected by mNGS swabs but excluded just like it was in the culture results?

214: please specify are you using Enterobacter to describe only the genus Enterobacter species (which looks like maybe on 11 detections) or all Enterobacteriaceae family (including E.coli,

klebsiella, etc) which would get to 23 detections and what I think you are describing. This needs to be corrected throughout paper as Enterobacter (genus) is not the proper name for this family of bacteria and is misleading and confusing.

232: is disease type referring to cancer type? Please clarify

242: would clarify the definition of EBV infection. Is this detection of EBV virus by PCR? Did EBV infection result in worse outcomes? EBV usually associated with increased mortality risk. Would also include culture colonization and outcome measure HR for OS, NRM, PFS. If no difference with culture colonization please state so.

288: would clarify utility of CRE detection by mNGS. Hard to understand the numbers here, but I think it sounds like mNGS only detected one CRE. The rest were identified by routine methods? Was any CRE detected by rectal swab, which is the routine method for detecting CRE colonization.

Discussion:

337: would clarify that you are likely excluding non-pathogenic organism detection.

359: would clarify by detection using mNGS

Discussion would benefit from further exploration into potential pathophysiology of link between colonization and thrombosis and other clinical examples showing this link and also if this thrombosis lead to worse outcomes. Further analysis into outcomes would be beneficial as well. Colonization and delayed platelet engraftment does not have an obvious pathophysiology explanation and without statistical significance would consider removing. Additionally, I think some exploration and/or suggestions into potential interventions. Ie if colonization was detected or CRE could changes in antibiotic prophylaxis, empiric antibiotics, delay in transplant and/or efforts to decolonize as efforts to improve outcomes. Similarly, some basic data on if these patients are placed on routine antibiotic prophylaxis or what empiric treatment for neutropenic fever looks like.

Dear Editors and Reviewers:

Thank you very much for your thorough review and valuable feedback on our manuscript. We have carefully considered each of your suggestions and have made the necessary revisions accordingly. Below, we provide our point-by-point responses to your comments:

Reviewer1

- It's not clear why Enterobacter colonies (from culture) were only inoculated onto meropenem chromogenic agar. Was this just to evaluate resistance to meropenem on samples that would otherwise still be considered culture positive for Enterobacter or are samples that were successfully cultured on the meropenem chromogenic agar the only samples classified as being colonized? If it's the latter, and only patients with cultures growing on meropenem were considered "colonized", I wonder if this might be missing out on positive cultures for meropenem-susceptible Enterobacter and possibly explain some of the differences between culture and mNGS results. The mNGS results would account for all Enterobacter in the sample, not just CRE, so comparing all Enterobacter sequence data to just CRE culture data could possibly explain some of the differences.

Response: Thank you very much for pointing out the ambiguity in our manuscript. In the "Testing Methods" section, we described the oropharyngeal swab culture process; however, we did not clearly define the criteria for identifying Enterobacteriaceae colonization via culture, which may have led to confusion. When using culture methods to identify oropharyngeal colonization, we first inoculated oropharyngeal swabs onto Columbia agar plates. If Enterobacteriaceae were detected, the sample was classified as positive for oropharyngeal Enterobacteriaceae colonization. Subsequently, carbapenem resistance was assessed to identify patients with carbapenem-resistant Enterobacteriaceae (CRE) colonization. We have revised the "Culture" subsection under "Testing Methods" to clarify this definition.

- Were any negative control swabs processed alongside patient swabs? Or were there any other processing controls in place to make sure that there weren't reagent contaminants that could potentially mediate some of the differences in culture vs mNGS results?

Response: Your concerns are entirely valid. In clinical practice, distinguishing between colonization and contamination is crucial. To exclude environmental and

laboratory contaminants, an external negative control (nuclease-free water) was included in each mNGS assay run in our study. Moreover, the patients included in this study differ from the general patient population in important ways. All enrolled patients were undergoing hematopoietic stem cell transplantation (HSCT) in a controlled cleanroom environment. These cleanrooms adhere strictly to air quality control, environmental management, personnel protection protocols, disinfection procedures, and infection control measures to minimize the risk of infection. Additionally, patients in cleanrooms undergo rigorous oral hygiene protocols, including mandatory mouth rinsing before sample collection and strict adherence to standardized sampling procedures. The “Sample Collection” section of the manuscript provides a detailed description of these procedures.

- Were there any alignment thresholds/coverage levels identified for classifying a mNGS sample as "colonized"? A quick explanation of which criteria were used to identify a sequenced sample as colonized vs a sample that might have had a small amount of spurious or misclassified reads but did not actually contain the bacteria would be helpful.

Response: Your suggestion is very reasonable. Firstly, we performed quality control on the mNGS sequencing data: adapter sequences at the 3' and 5' ends of the reads were trimmed using fastp (version 0.20.0), and reads shorter than 50 bp or with an average base quality score below 20 were removed, retaining only high-quality sequences. The reads were then aligned to the host DNA sequence using BWA software (version 0.7.17), and contaminant reads with high alignment similarity were removed. In our study, none of the patients exhibited clinical symptoms of infection at pharyngeal swab sampling; therefore, any positive results reported by mNGS were classified as colonizing pathogens. The standardised reads for genus and species were defined as the number of reads normalised to a data volume of 20 million. When multiple species are detected within the same genus, the species with the highest number of raw reads was used in the analysis. For most bacteria and virus, a positive result was determined when at least 3 stringently mapped read number (SMRN) are detected at the genus or species level. Owing to the thick fungal cell wall, which makes extracting sufficient DNA difficult, the threshold for a positive fungal result

was defined as detecting at least 3 reads when the depth ratio was ≥ 0.5 or the Shannon-Index was ≥ 0.75 . Common colonized bacteria and contaminating microorganisms, including *Rothia*, *Corynebacterium*, *Actinomyces israelii*, *Neisseria* and *Streptococcus* etc., were considered as microecological bacteria in our study.

Reviewer2

Throughout the paper, "Enterobacter" needs to be corrected to correctly identify the order of enterobacterales or family Enterobacteriaceae. Currently the manuscript uses Enterobacter to describe a group of organisms incorrectly, when Enterobacter only refers to the genus.

Overall, the methods and analysis in this study are sound but can use some clarity in how pathogens were identified or excluded from analysis. For example, in oropharyngeal and rectal swabs there should always be microorganisms that would be detected as normal flora and are often routinely excluded in the microbiology lab in clinical reporting. Similarly, mNGS in a non sterile space like the oropharynx should have significantly more pathogens detected than reported in this manuscript, and the manuscript would benefit from how these non-pathogens were excluded. Additionally, there should have been significantly more enterobacterales identified in rectal swabs (unless this is only detecting CRE). The inclusion of rectal swab data adds little to this manuscript other than to show routine screening is not very effective, but does not relate well to oropharyngeal mNGS. Additionally figure 2C is difficult to read which column matches with which organism. The Fungi identified by mNGS are interesting and would be good to comment on them even though small number. Did this patient have poor outcomes or any fungal infections. Phenotypic data is described in the methods but not described elsewhere, would be interesting if any analysis if any resistance identified associated with outcomes otherwise would consider removing.

Figure 3 and 4 is a lot of data to review and not addressed well in the manuscript. It should either be simplified or explained further in the manuscript. Additionally, I am not sure Kaplan-meier curves add much to the overall hazard ratios described in the manuscript.

Response : We greatly appreciate your sincere suggestions regarding our study. Based on your feedback, we have revised the manuscript and made adjustments to the

figures. Considering the potential impact of intestinal microbiota on rectal swab samples and the cost of mNGS, we only performed culture to screen for perianal colonization in our study. Due to the lower sensitivity of cultures, the detection rate of perianal colonization was not high, and the specific microbial species detected are listed in Figure 1. In our study, only six cases of oropharyngeal fungal colonization were detected through mNGS, and the detection of fungi by mNGS is indeed noteworthy. Although the detection rate of oropharyngeal fungal colonization is low, follow-up results revealed that one patient, whose oropharyngeal swab tested positive for *Pneumocystis jirovecii*, developed *Pneumocystis jirovecii* pneumonia (PJP) after allo-HSCT and subsequently died. As this is a prospective study, there are limitations due to sample size. Therefore, the number of cases of CRE is relatively low, making it difficult to establish a direct correlation between antimicrobial resistance phenotypes and outcomes. However, we did confirm in the results section that patients with CRE colonization had poorer prognoses.

Specific Comments:

Abstract:

29-30: I do not consider colonization screening as routine care. Is this screening for MRSA, CRE colonization? Any studies to support this or the benefits to justify screening?

Response: The "Introduction" describes the importance of colonization screening for HSCT patients, highlighting its significant role in predicting and preventing infections during the transplant process. Therefore, we conduct weekly oropharyngeal and anal swab cultures on HSCT patients to screen for colonizing bacteria. We have adopted this process as the necessary approach for this study.

30-32: I do not believe mNGS has been widely adopted in clinical practice. It has a small niche use in rare, atypical or uncultivable infections or in those already receiving antibiotics where culture growth is unlikely.

Response: I completely agree with your view. Due to the high cost of mNGS, it has not been widely applied in clinical practice. However, mNGS plays a crucial role in the diagnosis of complex and challenging infections. I have revised the expression in

the manuscript accordingly.

38-39: This is not unexpected but would specify that organisms were detected not colonization rate. As every individual is colonized with organisms in their oropharyngeal space as it is a nonsterile site. There colonization rate is 100% but you are changing the rate of detection based on diagnostic modality.

Response : I have made revisions to correct the inaccurate expressions in the manuscript.

Introduction:

73-79: would elaborate on this study further. Would consider adding what the rate of pathogen culture detection is in a normal host oropharyngeal culture for comparison. The detection rate does not highlight the clinical importance of screening. I would include any differences in outcomes based on screening detection which would highlight clinical importance.

77: Klebisella and Enterobacter are Enterobacteriaceae species, not Enterobacter species 80: I don't think there are any clinically relevant oropharyngeal organisms that take weeks to culture.

85-87: yes mNGS is being used in several clinical sequencing tasks, but is still not clinically superior to cultures for pathogen detection in most cases.

91-93: again mNGS is not widely used in routine clinical medicine but is occasionally used as an adjunct to routine culture, antibody, antigen, and/or PCR detection.

*94-97: I agree with these limitations but would also include significantly increased costs compared to routine methods, and specialized trained personnel, and often turnaround time is still equal if not longer than routine **detection methods**.*

Response : In response to your comments on this section, I have made the necessary revisions and additions in the manuscript.

Methods:

116: not sure how much of this routine neutropenic hygiene protocol is necessary. Would consider adding if patients were on specific antibiotic/antifungal prophylaxis for neutropenia.

139: would specify streptococcus species. While many are normal flora, several species are pathogen and even normal flora strep species can lead to bacteremia especially in neutropenic patients with mucositis.

159-167: this phenotypic testing should be included in the culture methods, including PCR sequencing as this is part of routine culture methods and not mNGS. Then would add how any resistance genes may have been detected using mNGS.

Response : In response to your comments on this section, I have made the necessary revisions and additions in the manuscript.

Results:

194: clarify colonization rate as colonization with pathogenic organisms.

197-198: Please clarify the rectal colonization rate. As all of these organisms can be considered normal flora of rectal swabs and I would suspect a much higher rate of colonization with at least one of these organisms in stool samples if not 100%. Are all of these CRE organisms? If so should specify so and would be more clinically relevant.

204: does the bacterial mNGS detection rate include only clinically relevant bacterial detections? Were many species excluded or removed. Figure 2C makes me think that only certain species were included in the mNGS results. For example mNGS does not list any strep species but I'm sure some strep was detected by mNGS swabs but excluded just like it was in the culture results?

214: please specify are you using Enterobacter to describe only the genus Enterobacter species (which looks like maybe on 11 detections) or all Enterobacteriaceae family (including E.coli, klebsiella, etc) which would get to 23 detections and what I think you are describing. This needs to be corrected throughout paper as Enterobacter (genus) is not the proper name for this family of bacteria and is misleading and confusing.

232: is disease type referring to cancer type? Please clarify

242: would clarify the definition of EBV infection. Is this detection of EBV virus by PCR? Did EBV infection result in worse outcomes? EBV usually associated with increased mortality risk. Would also include culture colonization and outcome measure HR for OS, NRM, PFS. If no difference with culture colonization please state so.

288: would clarify utility of CRE detection by mNGS. Hard to understand the numbers here, but I think it sounds like mNGS only detected one CRE. The rest were identified by routine methods? Was any CRE detected by rectal swab, which is the routine method for detecting CRE colonization.

Response: In response to your comments on this section, I have made the necessary revisions and additions in the manuscript.

Due to the fact that only cultures were tested for the anal swabs, the detection rate was relatively low. Based on rectal swab cultures, four cases of CRE were detected, three

of which also had CRE colonization in the oropharynx. Since mNGS data was lacking for anal swabs in our study, it was not a primary focus for further discussion.

In Figure 2C, we have only listed the top 10 pathogens based on detection rate. Certain microorganisms, such as Streptococcus and Neisseria were listed in the supplementary report and are considered normal microbiota of the oropharynx.

I have revised the term "Enterobacter" throughout the manuscript.

The "Disease types" section is consistent with the "Underlying diseases" in SupplementaryTable2.

CMV or EBV infection was defined as a CMV DNA viral load >100 copies/mL or an EBV DNA viral load >100 copies/mL at any point after HSCT based on at least one measurement. In our study, 18 patients experienced EBV activation within 100 days post-transplant, and none of these patients died within this period. However, as the follow-up time extended, six patients died. Although the cause of death was not directly linked to EBV infection, the association between EBV infection and an increased risk of mortality certainly warrants further investigation. We did not find differences between culture colonization and outcomes based on the competing risk curves and K-M survival curves, as explained in Lines 259-263, and therefore no further multivariable analysis was conducted.

Regarding the diagnostic value of mNGS in CRE, the description in the manuscript may have been somewhat ambiguous. Of the five patients with throat colonization, three were detected by mNGS, and two were detected by conventional methods.

Discussion:

337: would clarify that you are likely excluding non-pathogenic organism detection.

359: would clarify by detection using mNGS Discussion would benefit from further exploration into potential pathophysiology of link between colonization and thrombosis and other clinical examples showing this link and also if this thrombosis lead to worse outcomes. Further analysis into outcomes would be beneficial as well. Colonization and delayed platelet engraftment does not have an obvious pathophysiology explanation and without statistical significance would consider removing. Additionally, I think some exploration and/or suggestions into potential

interventions. Ie if colonization was detected or CRE could changes in antibiotic prophylaxis, empiric antibiotics, delay in transplant and/or efforts to decolonize as efforts to improve outcomes. Similarly, some basic data on if these patients are placed on routine antibiotic prophylaxis or what empiric treatment for neutropenic fever looks like.

Response: In response to your comments on this section, I have made the necessary revisions and additions in the manuscript.

We tried our best to improve the manuscript and uploaded the revised version of the manuscript. We appreciate for Editors/Reviewers' work earnestly, and hope the correction will meet with approval. Once again, thank you very much for your comments and suggestions.

Re: Spectrum00028-25R1 (Clinical Value of Metagenomic Next-Generation Sequencing in Screening Oropharyngeal Colonization in Patients Undergoing Allogeneic Hematopoietic Stem Cell Transplantation: A Prospective Observational Study)

Dear Prof. Xiaojin Wu:

Your manuscript has been accepted, and I am forwarding it to the ASM production staff for publication. Your paper will first be checked to make sure all elements meet the technical requirements. ASM staff will contact you if anything needs to be revised before copyediting and production can begin. Otherwise, you will be notified when your proofs are ready to be viewed.

Sincerely,
Po-Yu Liu
Editor
Microbiology Spectrum

Reviewer #1 (Comments for the Author):

I have no additional comments or suggestions; the revisions related to my comments have been adequately addressed.

Reviewer #2 (Comments for the Author):

Thank you for this well written and interesting manuscript. I have reviewed prior reviewer comments and feel that all questions and concerns have been adequately corrected and addressed in the revised manuscript.

The manuscript nicely demonstrates the added benefit from routine culture screening with mNGS screening of oropharyngeal samples with increased detection of pathogens and the increased risk when enterobacteriaceae and CRE is detected.